# Phototoxic Potential of Different DNA Intercalators for Skin Cancer Therapy: In Vitro Screening

**DOI:** 10.3390/ijms24065602

**Published:** 2023-03-15

**Authors:** Thais P. Pivetta, Tânia Vieira, Jorge C. Silva, Paulo A. Ribeiro, Maria Raposo

**Affiliations:** 1CEFITEC, Department of Physics, NOVA School of Science and Technology, Universidade NOVA de Lisboa, 2829-516 Caparica, Portugal; 2Laboratory for Instrumentation, Biomedical Engineering and Radiation Physics (LIBPhys-UNL), Department of Physics, NOVA School of Science and Technology, Universidade NOVA de Lisboa, 2829-516 Caparica, Portugal; 3CENIMAT/I3N, Departamento de Física, Faculdade de Ciências e Tecnologia, Universidade Nova de Lisboa, 2829-516 Caparica, Portugal

**Keywords:** photodynamic therapy, photosensitizer, skin cancer, reactive oxygen species

## Abstract

Photodynamic therapy is a minimally invasive procedure used in the treatment of several diseases, including some types of cancer. It is based on photosensitizer molecules, which, in the presence of oxygen and light, lead to the formation of reactive oxygen species (ROS) and consequent cell death. The selection of the photosensitizer molecule is important for the therapy efficiency; therefore, many molecules such as dyes, natural products and metallic complexes have been investigated regarding their photosensitizing potential. In this work, the phototoxic potential of the DNA-intercalating molecules—the dyes methylene blue (MB), acridine orange (AO) and gentian violet (GV); the natural products curcumin (CUR), quercetin (QT) and epigallocatechin gallate (EGCG); and the chelating compounds neocuproine (NEO), 1,10-phenanthroline (PHE) and 2,2′-bipyridyl (BIPY)—were analyzed. The cytotoxicity of these chemicals was tested in vitro in non-cancer keratinocytes (HaCaT) and squamous cell carcinoma (MET1) cell lines. A phototoxicity assay and the detection of intracellular ROS were performed in MET1 cells. Results revealed that the IC_50_ values of the dyes and curcumin in MET1 cells were lower than 30 µM, while the values for the natural products QT and EGCG and the chelating agents BIPY and PHE were higher than 100 µM. The IC_50_ of MB and AO was greatly affected by irradiation when submitted to 640 nm and 457 nm light sources, respectively. ROS detection was more evident for cells treated with AO at low concentrations. In studies with the melanoma cell line WM983b, cells were more resistant to MB and AO and presented slightly higher IC_50_ values, in line with the results of the phototoxicity assays. This study reveals that many molecules can act as photosensitizers, but the effect depends on the cell line and the concentration of the chemical. Finally, significant photosensitizing activity of acridine orange at low concentrations and moderate light doses was demonstrated.

## 1. Introduction

Photodynamic therapy (PDT) has been widely explored in recent decades for the treatment of diseases such as skin disorders and some cancer types [1,2]. PDT is a minimally invasive therapy with local action [3], the efficacy of which depends on the presence of photosensitizer (PS) molecules, oxygen and light. For skin applications such as in skin squamous cancers and melanomas, PDT can be more convenient than invasive or systemic treatment [4,5]. The success of PDT relies on the combination of these elements, which depends on the formation of reactive oxygen species (ROS) that can be produced by two types of PDT [6]. In type I, the activated photosensitizer acts on substrates such as the biological molecules that react with oxygen-generating ROS. When the photosensitizer can transfer energy to molecular oxygen (^3^O_2_), generating singlet oxygen (^1^O_2_), the PDT is considered type II. In both paths, ROS with high oxidizing power are generated, leading to cell death through damage to biomolecules [7]. DNA is among the biomolecules that can be subjected to damage and is the main target for anticancer drugs [8]. Due to the short lifetime of reactive species, they act essentially at the local level, where they were created, showing the advantage of PDT as a therapy with local action [9]. 

The first generation of photosensitizers was those of the porphyrin type [10,11]. In fact, derivative molecules from hematoporphyrin have been used since the middle of the 20th century. PDT using porphyrins as photosensitizers has proven to be efficient in cancer therapy and a promising sensitizer to deal with microbial infections [2,7]. However, the wavelength able to activate porphyrin and hematoporphyrin photosensitizers is usually too short, which hinders light penetration in tissues. Moreover, after activation, the half-life is too long, which can cause severe phototoxicity. Later, Photofrin^®^, which is composed of monomers, dimers and oligomers of hematoporphyrin derivative, exhibiting a longer wavelength of activation was approved [11]. A second generation of photosensitizers was developed through alterations in the porphyrin core or in peripheral sites to improve photosensitivity. Phthalocyanines are examples of second-generation photosensitizer analogues from porphyrins [11]. There are more than 400 molecules with photosensitizing potential, including dyes, natural products and many other chemicals [7,10]. 

Whether of natural or synthetic origin, dyes are substances able to provide color and are usually water-soluble. There are several applications for these molecules, such as in textiles, foods, cosmetics and medicine [12]. In 1900, Oscar Raab left the protozoan *Paramecium caudatum* in contact with the dye acridine orange and observed a toxic effect after sunlight exposure [13]. Phthalocyanines are synthetic dyes extensively investigated for use in many types of cancer therapies, namely ovarian carcinoma, melanoma, liver carcinoma and lung carcinoma [14,15,16]. Methylene blue is another synthetic dye that has attracted interest in studies of PDT as an antimicrobial and for cancer treatment [10]. Dos Santos et al. [17] investigated the potential selectivity to tumor cells and the potential use of MB in photodynamic therapy in human breast cancer cells. Treatment with low MB concentrations such as 2 μM and 20 μM and a dose of 4.5 J/cm² were showed to be determinant in causing massive death of cancer cells, while non-malignant cells were more resistant.

Since antiquity, natural products such as herbs or extracts have been employed to treat several diseases [18]. There are several classes of molecules with PS properties, such as anthraquinones, curcuminoids, xanthenoids, etc. For example, hypericin is a natural molecule that belongs to the anthraquinones class and is the first non-porphyrin-like molecule to be recently introduced in clinical trials [19]. Curcumin application in photodynamic therapy for cancer is has also been widely investigated through its incorporation in nanocarriers [20,21,22]. De Matos et al. [23] studied a curcumin nanoemulsion in two different cell lines derived from uterus carcinoma and observed a phototoxicity of 93% for Ca Ski and 83% for SiHa cells using 20 μM of encapsulated curcumin. Studies involving the combination of the flavonoid quercetin with PDT revealed reduced cell viability of cervical adenocarcinoma and breast carcinoma cells [24,25]. Furthermore, some complexes have been investigated, such as the agent chelators 2,2′-bipyridyl and 1,10-phenanthroline, which were studied in metallic complexes and presented toxicity in cervical adenocarcinoma HeLa cells [26]. 

Therefore, as many classes of molecules can act as photosensitizers, in this work, we analyzed the phototoxic potential of some DNA-intercalating agents: the dyes methylene blue, acridine orange and gentian violet; the natural products curcumin, quercetin and epigallocatechin gallate; and the chelating compounds neocuproine, 1,10-phenanthroline and 2,2′-bipyridyl [27,28,29,30,31,32,33,34,35]. We analyzed this set of molecules with respect to their potential as photosensitizers and, consequently, their use in PDT. Results revealed that MB and AO molecules present a significant phototoxic potential in MET1 and WM983b cell lines, which is an unprecedent result for these cell lines.

## 2. Results and Discussions

### 2.1. Cytotoxicity in HaCaT and MET1 Cells

To study cell viability in the presence of different DNA-intercalating agents, HaCaT and MET1 cells were exposed to a wide range of concentrations (1.25–320 μM) of those agents for 24 and 48 h. Figure 1 and Figure 2 show the relative cell populations of HaCaT and MET1 cells, respectively, treated with dyes (methylene blue, acridine orange and gentian violet), natural compounds (quercetin, curcumin and epigallocatechin gallate) or chelating molecules (neocuproine,1,10- phenanthroline and 2,2′-bipyridyl). The values of relative cell viability as a function of PS concentration allowed for the calculation of the half-maximal inhibitory concentration (IC_50_) for the different chemicals tested. IC_50_ is usually employed to evaluate a substance efficacy and the concentration able to inhibit cell growth [36]. The IC_50_ values are displayed in Table 1; however, for some compounds, only a lower limit could be evaluated.

In general, dyes show low IC_50_ values of less than 20 µM in both non-cancer and cancer cell lines (Figure 1A and Figure 2A). However, results show that there is a slightly lower value of IC_50_ for the dyes in non-cancer keratinocytes compared to cancer keratinocytes. For natural products, the results show higher IC_50_ values compared to the dyes. In addition, the natural products were less toxic for HaCaT cells (Figure 1B) than for MET1 cells (Figure 2B). Many researchers have demonstrated the selectivity of natural products, which were reported more toxic to cancer cells than to non-cancer cell lines [37,38,39], as in the work of Srivastava [30] regarding the selectivity of the polyphenol quercetin acting against cancer cells. Taking quercetin as an example, there are indications that it causes cell death through apoptotic signaling pathways such as via the tumor suppressor p53 protein [30,40,41]. The chelating agents showed lower IC_50_ values for the non-cancer cell line (Figure 1C) relatively to the values presented for the cancer keratinocytes (Figure 2C).

The high cytotoxicity of dyes for the MET1 cells is a promising result, considering their use as anticancer drugs. However, in the pursuit of photosensitizers, the light effect should also be considered. MB, AO and GV are very toxic for MET1 cells, exhibiting IC_50_ values of 14.2 µM, 15.1 µM and 3.9 µM after an exposure time of 24 h and 4.2 µM, 10.3 µM and 1.2 µM after an exposure time of 48 h, respectively. Clearly, GV was the most toxic agent among the dyes, which is a challenge, considering that it would be interesting to observe a difference in cell viability when applying light irradiation for the evaluation of the phototoxic potential.

The molecule curcumin has been widely studied regarding its anticancer properties [42,43] and showed low IC_50_ values of 26 µM and 7.2 µM in squamous cells carcinoma at 24 h and 48 h, respectively. Moreover, according to the literature [44,45,46,47], EGCG and quercetin are also natural candidates for anticancer therapy. However, in this work, the achieved IC_50_ values are not very impressive compared with those of other molecules. This same perception occurred for the chelating agents, which also presented IC_50_ values greater than 100 µM in the 24 h experiment.

### 2.2. Phototoxicity in MET1 Cells

To evaluate the phototoxic potential of the DNA-intercalating agents for cancer cells, irradiation studies were carried out using MET1 cells. The range of concentration was reduced, since some molecules, mainly the dyes, induced more than 90% cytotoxicity at concentrations above 40 µM, as previously observed in the cytotoxicity experiments. The DNA-intercalating molecules, as potential photosensitizers, were tested with the MET1 squamous cell carcinoma line and submitted to different light sources to evaluate their phototoxic effect in comparison with samples kept in the dark without irradiation (control). 

Figure 3A–C exhibit the cell viability results for the dyes MB, AO and GV, respectively, submitted to irradiation or not. For this assay, cells were treated for 24 h, then irradiated with different wavelengths or kept in the dark for the non-irradiated samples. A statistical comparison between the treated MB cells shows that all the light wavelengths affected the cells’ viability compared to the control cells kept in the dark. Cells treated with MB and irradiated with red light (640 nm) showed the lowest IC_50_ value of 3.8 μM, which is almost three times lower than that of the control, as exhibited in Table 2. Irradiation with the 254 nm wavelength light also showed great phototoxicity, with an IC_50_ value of 4.3 μM. This was expected, since MB has three main characteristic bands: two in the UV region, with maximum absorbance at 250 nm and 300 nm, and one in the visible region, with maximum of absorbance at 665 nm [48]. For this reason, the photodynamic potential of MB to damage cancer cells has attracted the attention of researchers. For example, Kofler et al. [49] studied the effect of MB in head and neck squamous cell carcinoma, while Dos Santos et al. [17] investigated the effect in breast cancer cells and reported that MB-mediated PDT caused significant cancer cell death.

The effect of the AO (Figure 3B) on cell viability is strongly dependent on the light wavelength. The 457 nm light induced a viability below 10% on MET1 cells for all investigated concentrations. This is evident when analyzing the IC_50_, which shows calculated values of 11.5 µM, 6.6 µM, 4.1 µM and 7.1 µM for dark wavelengths and 640 nm, 583 nm and 254 nm for light wavelengths. For the 457 nm irradiation, the viability is even lower, and the experiment was repeated using concentrations between 0.3 and 10 µM for the calculation of the IC_50_ value (Appendix A). The obtained IC_50_ value was 0.4 µM. These results are in agreement with the AO spectrum, which exhibits characteristic peaks in the UV region around 230 nm, 270 nm and 290 nm [50]; in the visible region at 490 nm (maximum absorbance) due to the AO molecule (monomer); and a shoulder at 470 nm that can turn into a peak with an increase in concentration, possibly due to AO dimers [51]. Like AO, several molecules with intense absorption peaks in the blue wavelength region (Soret peak) are described in the literature as very efficient PS [7]. Osman et al. [52] investigated the photosensitizing potential of AO for glioblastoma and, using a low concentration of 0.001 mg/mL, verified a significant decrease in cell count. The potential application of AO-PDT for bladder cancer therapy was studied by Lin et al. [53], reported significant cell death using low AO concentrations and blue light irradiation.

Cells treated with GV and irradiation were also affected by light, with the exception of blue light (457 nm), as shown in Figure 3C. In particular, a decrease in cell viability was observed under irradiation at 583 nm, resulting in an IC_50_ value of 1.0 µM. However, as previously shown in cytotoxicity studies, this dye presents a high toxicity towards the MET1 cell line. The relative cell viability for the dark condition at the lowest concentration of 1.25 µM was below 60%, while for the other PS, we obtained at least 90% cell viability under the same condition.

In the concentration range of 1.25 µM to 40 µM, QT showed significant results at 40 µM under irradiation with 583 nm, 457 nm and 254 nm wavelength light (Figure 3D). For EGCG, no phototoxicity was observed in the investigated range. EGCG is a molecule that, depending on the concentration, can fulfill a dual role as a pro-oxidant and antioxidant [54,55]. At nanomolar and low micromolar concentrations, EGCG can present an antioxidant effect [54], exhibiting a pro-oxidant effect at higher concentrations, as in the study by Zhou et al. [56], in which EGCG was reported to play a different role as a pro-oxidant above 100 μM. To calculate the IC_50_ values corresponding to these molecules, assays were repeated using concentrations up to 320 µM (Appendix A), the results of which are presented in Table 2. One can observe that for QT and EGCG, the IC_50_ values were, in general higher than those of the dark control. For QT, the control presented an IC_50_ of 230 µM and was lower when irradiated at 254 nm, with an IC_50_ value of 205 µM. For EGCG, all IC_50_ values obtained in the phototoxicity studies (Figure 3E) were above that of the control (190 µM). These compounds are often used in combination with other agents for applications in PDT [24,25,57,58], as seen in the research reported by Mun et al. [57], who demonstrated that PDT using Radachlorin and EGCG improved the antitumor effects on TC-1 tumor cells both in vitro and in vivo.

Cell viability studies in the presence of curcumin and irradiation with light at wavelengths of 640 nm, 583 nm, 457 nm and 254 nm showed statistically different results for the populations of the experimental conditions in comparison with the dark control, with IC_50_ values below 11.5 µM (Figure 3F). As curcumin presents a characteristic absorbance band in the visible region at 430 nm [59], a strong effect is expected to be observed for the 457 nm light. An IC_50_ of 10.7 µM was observed, which is lower than that of the control, with an IC_50_ of 15.5 µM. Fadeel and collaborators [60] investigated the encapsulation of curcumin in a PEGylated lipid carrier to be used in photodynamic therapy of a human skin cancer cell line. They demonstrated that the cell survival decreased even further when using curcumin suspension, which was boosted when using encapsulated curcumin.

Studies using lower concentrations of neocuproine (1.25–40 µM) showed that cell viability slightly increased as the concentration increased (Figure 3G). Due to this behavior, the IC_50_ value cannot be calculated. However, similarly to GV, this compound shows a cell viability below 70%, even under dark conditions. For 1,10-phenanthroline (Figure 3H), it is possible to observe significant differences in cell populations between the dark control and the irradiated conditions at 5 µM and 10 µM. The application of chelators such as phenanthroline in PDT is usually associated with incorporation into a metallic complex, as revealed by Al Hageh et al. [61], who reported that synthetized bis-phenanthroline complex was able to generate DNA damage after irradiation due to successful ROS production.

The chelating agent 2,2′-bipyridyl did not show phototoxicity in the concentration range exhibited in Figure 3I; therefore, experiments were repeated with concentrations up to 320 µM (Appendix A) for determination of the IC_50_. In this case, a decrease in the IC_50_ values of the irradiated samples occurred at light wavelengths of 583 nm and 457. Similarly, to 1,10-phenanthroline, some studies showed the efficient photosensitizing activity of 2,2′-bipyridyl in metallic complexes [62,63]. Therefore, the association of these molecules in complexes is an interesting approach to be explored.

In summary, some of the investigated compounds, such as the dyes, presented phototoxicity at low concentrations, while natural compounds, with the exception of curcumin, presented phototoxicity above 150 µM. Considering the results obtained with MET1 cells, methylene blue and acridine orange at low concentrations are worthy of further studies with the melanoma cell line WM983b and evaluation of intracellular ROS production by MET1 cells.

### 2.3. Intracellular ROS Production in MET1 Cells

As the phototoxicity results showed that both MB and AO caused a significant decrease in the IC_50_ values when 640 nm and 457 nm wavelength light was applied, fluorescence studies were directed to these molecules using low concentrations known to have no effect on cell viability in the absence of irradiation. 

DHE oxidizes in the presence of the superoxide radical (O_2_^•−^), making it the selected probe to evaluate ROS production by fluorescence microscopy due to the formation of stable products with red fluorescence, such as 2-hydroxiethidium (2-OH-E^+^) and ethidium (E^+^), the latter as result of non-specific oxidation such as with the hydroxyl (^•^OH) radical or hydrogen peroxide (H_2_O_2_) [64]. As DHE presented a red fluorescence and its spectrum of excitation/emission was not affected or interfered with by our treatments with the dyes, DHE was chosen as probe to detect the reactive oxygen species.

Fluorescence images of cells treated with MB in the presence or in the absence of light are shown in Figure 4. It is possible to observe a weak fluorescence intensity in the cells of the control (0 µM, without MB) and in the cells treated with MB but no light. However, we can observe a slight increase in fluorescence intensity at the concentrations of 2.5 µM and 5 µM in irradiated cells relative to non-irradiated cells. This may indicate modest ROS production after irradiation. Cells treated with AO and submitted to irradiation and non-irradiated cells treated with AO showed a slight increase in fluorescence intensity with the increase in AO concentration (Figure 5) compared to the cells treated with MB (Figure 4). Moreover, a clear difference between the fluorescence intensity of cells submitted to irradiation compared to those without the irradiation, with the increase in AO concentration, is notorious. Thus, we can infer the production of ROS after irradiation.

Figure 6 depicts the quantification of red fluorescence intensity measured for each concentration of MB (Figure 6A) and AO (Figure 6B) using the ImageJ software. The study with MB (Figure 6A) shows an integrated density of fluorescence intensity around 1.1 × 10^6^ for the non-irradiated control (0 µM, without the dye). However, a modest increase in the integrated density for samples treated with 1.25 µM and 2.5 µM of MB was observed, followed by a decrease for the highest concentration, in agreement with the fluorescence images exhibited in Figure 4. One can speculate that through this method, it was possible to detect a small amount of ROS, mainly at lower concentrations of dyes. However, it was clear that MB presented phototoxicity in this cell line at these concentrations in such a way that the employed probe may not have been the most appropriate in this case; therefore, other ROS probes should be tested.

The fluorescence intensity for cells treated with AO is displayed in Figure 6B. One can observe an integrated density value around 1.1 × 10^6^ for the control without AO. With an increase in AO concentration, the integrated density intensity is seen to rise for samples with and without light irradiation. However, the greater fluorescence intensity is associated, without any doubt, with the irradiated samples. In comparison to the irradiated control (0 µM, without AO), cells treated with 5 µM AO and submitted to 457 nm irradiation presented an increase in fluorescence about 2.6-fold. Comparing non-irradiated cells treated with 5 µM AO and the non-irradiated control (0 µM, without AO), there is an increase about 1.8-fold in the fluorescence intensity. AO can accumulate in acidic vesicular organelles and is often employed in fluorescence to investigate autophagy [65]; therefore, as at these concentrations, AO is not toxic, the mild red fluorescence of cells without light irradiation may indicate an autophagy process. Apart from the uncertain reason as to why AO creates a red fluorescence in the cells, it should be reinforced that the DHE detection method was able to confirm the ability of AO to induce the production of ROS.

### 2.4. Cytotoxicity and Phototoxicity in WM983b Cells

To evaluate the possibility of using the same promising photosensitizers—the dyes MB and AO—for other types of cancers, studies were performed with a melanoma cell line. Figure 7 shows the viability study of WM983b cells treated for 24 or 48 h with the dyes MB and AO in the concentration range of 1.25 µM to 320 µM. The IC_50_ values of the dyes were 18.0 µM and 6.1 µM for MB and 25.2 µM and 6.6 µM for AO, in the 24- and 48-h experiments, respectively. Compared to non-cancer and cancer keratinocyte cell lines, the melanoma cells revealed higher resistance to these compounds. However, they are clearly cytotoxic to this cell line, as observed at concentrations above 10 µM of MB and 20 µM of AO, i.e., concentrations at which the relative cell population falls below 70%. Melanoma cells rapidly develop resistance to treatment through various molecular mechanisms [66], so it is expected that melanoma cells would be more resistant to PDT than other cancer cells.

For the cells treated with MB, phototoxicity results show IC_50_ values of 2.1 µM and 7.3 µM for irradiated and non-irradiated samples, respectively (Figure 8A). In accordance with previous results with the squamous carcinoma cell line, the IC_50_ values for AO in melanoma cell line were 1.1 µM and 17.5 µM with and without irradiation, respectively. Once again, AO molecules caused more significant phototoxicity that that achieved with MB. In both cases, PDT is effective for this melanoma cell line at low dye concentrations.

## 3. Materials and Methods

### 3.1. Materials

Dulbecco’s Modified Eagle Medium (DMEM) and fetal bovine serum 2343 obtained from Biowest (Riverside, CA, USA). Penicillin–streptomycin and TrypLE™ express enzyme were obtained from Gibco, Thermo Fisher Scientific (Waltham, MA, USA). Resazurin was obtained from Alfa Aesar (Ward Hill, MA, USA). Dihydroethidium and Hoechst 33342 were obtained from Biotium (San Francisco Bay Area, CA, USA) and Molecular Probes, Thermo Fisher Scientific (Waltham, MA, USA), respectively. Methylene blue, acridine orange, gentian violet, curcumin, quercetin, EGCG, 1,10-phenanthroline, neocuproine and 2,2′-bipyridyl were obtained from Sigma Aldrich (St. Louis, MI, USA).

### 3.2. Cell Culture

The MET1 SCC cell line (human squamous cell carcinoma) was obtained from Ximbio (London, UK), HaCaT (immortalized human keratinocytes cell line) was obtained from Addexbio (San Diego, CA, USA) and WM983b (human metastatic melanoma cell line) was obtained from Rockland (Pottstown, PA, USA). Cells were maintained in Dulbecco’s Modified Eagle Medium (DMEM) supplemented with 10% (*v*/*v*) fetal bovine serum (FBS) for MET1 and HaCaT and 5% (*v*/*v*) for WM983b cells, and Pen–Strep (penicillin 100 U/mL and streptomycin 100 µg/mL). Cells were cultivated in an incubator (Sanyo MCO-19AIC-UV) at 37 °C in a 5% CO_2_ humidified atmosphere. 

Cytotoxicity studies were carried out using MET1 cells and HaCaT cells. Phototoxicity ROS production were determined using MET1 cells. Finally, WM983b cells were used to evaluate the cytotoxicity and phototoxicity of the most promising compounds.

### 3.3. Cytotoxicity

First, 96-well plates were seeded with HaCaT and MET1 cells at a density of 20,000 cells/cm² and incubated for 24 h. After that period, the culture medium was replaced by culture medium containing different compounds: MB, AO, GV, QT, CUR, EGCG, PHE, NEO and BIPY. The compounds were diluted in culture medium using a range of concentrations of each compound from 1.25 µM to 320 µM (in quadruplicate). The medium was aspirated from the 96-well plate and replaced by the samples diluted in medium. Cells cultured with complete medium and cells cultured with medium supplemented with 0.2% (*v*/*v*) or 10% (*v*/*v*) of DMSO were used as negative (NC), solvent (SC) and positive controls (PC), respectively. The plates were incubated for 24 and 48 h, followed by evaluation of cell viability using a colorimetric assay. Cells containing samples were washed with PBS supplemented with calcium and magnesium; then, resazurin solution diluted in DMEM at 0.02 mg/mL was added. The medium containing resazurin was also dispensed in wells without cells, which were used as references. After 3 h of incubation, the absorbance was measured in a microplate reader (ELX800UV, Biotek Instruments) at wavelengths of 570 nm and 600 nm. The corrected absorbance was proportional to cell viability. Propagation of uncertainties was used to calculate the combined standard uncertainty. For the cytotoxicity study using WM983b cells, plates were seeded at a density of 40,000 cells/cm², and cells were treated only with MB and AO.

### 3.4. Phototoxicity of MET1 Cells

For the phototoxicity studies, 96-well plates were seeded with MET1 cells at a density of 20,000 cells/cm², incubated overnight and treated for 24 h with MB, AO, GV, QT, CUR, EGCG, PHE, NEO and BIPY diluted in culture medium in a concentration range of 1.25 µM to 40 µM (in sextuplicate). Molecular spectra are displayed in Appendix A. Cells cultured with complete medium and cells cultured with medium supplemented with 0.2% (*v*/*v*) or 10% (*v*/*v*) of DMSO were used as negative (NC), solvent (SC) and positive control (PC), respectively. The plates were incubated for 24 h, followed by washing with PBS supplemented with calcium and magnesium. Afterwards, complete culture medium without phenol red was added, and the plates were irradiated from above and submitted to a dose of 2.5 J/cm² of irradiation emitted by different wavelength light sources coupled with a homemade irradiation system created in our laboratory. These light sources consisted of an equivalent 26 cm UV tubular lamp (254 nm) (Phillips TUV PLS 9W/2PHg) and a tricolor LED lamp (Led Grow Light Bulb E26/E27) with intensity peaks at 457 nm (blue light), 583 nm (yellow light) and 640 nm (red light) provided by 18, 18 and 28 LEDs, respectively, with dimensions of 3 × 3 mm^2^. The irradiance values of light at 254, 457, 583 and 640 nm wavelengths were of 16.7, 55, 45 and 25 W/m^2^, respectively. This experiment was performed in an incubator that was able to keep plates in a dark environment; therefore, the dark control plates were protected from any light source, as well as the others, with the exception of the irradiation that each one was submitted to. Plates were incubated for another 24 h, and finally, the resazurin assay was performed as previously described in Section 3.3. For the phototoxicity study using WM983b cells, plates were prepared at a cell density of 40,000 cells/cm², and cells were treated with MB and AO only.

### 3.5. Intracellular ROS Production

MET1 cells were seeded at a density of 20,000 cells/cm² in 24-well plates. After 24 h, cells were treated with MB and AO diluted in culture medium at low concentrations (1.25 µM, 2.5 µM and 5 µM). The medium culture was replaced by the prepared samples. Plates were incubated for 24 h; then, cells were washed with medium, and DMEM without phenol red was added. The plates were submitted to 2.5 J/cm² irradiation under red light at 640 nm for MB and 457 nm blue light for AO. Dihydroethidium (DHE) was used to evaluate ROS production. For this assay, the medium was removed after irradiation, and 20 µM of DHE diluted in medium was added. The plate was incubated for 20 min, washed with medium and incubated 20 min with Hoechst 33342 (5 µg/mL), which was used to stain the cells’ nuclei. After removing the excess dye, culture medium was added, and the cells were immediately observed using a Nikon Ti-S epifluorescence microscope. Cells were examined using a 40 × objective in random areas of 4 different wells under the same treatment condition. Images were analyzed using ImageJ software [67], and the intensity was measured according to integrated density values of red fluorescence.

## 4. Conclusions

In this work, the in vitro cytotoxicity of several DNA-intercalating agents, namely MB, AO, GV, CUR, QT, EGCG, NEO, PHE and BIPY, was analyzed in HaCaT (non-cancer keratinocytes) and MET1 (cancer keratinocytes). The achieved results allowed used to conclude that non-cancer keratinocytes are more sensitive to the tested agents than the cancer cell line. In general, dyes and curcumin were revealed to be more cytotoxic (IC_50_ below 30 µM) than the other tested compounds. Phototoxicity results showed that the IC_50_ values of the dyes and curcumin were very low (below 20 µM), while those of the natural products QT, EGCG and the chelating agent BIPY were above 100 µM for MET1 cells. The most promising molecules that considerably reduced their IC_50_ values were MB and AO when submitted to a 640 nm and 457 nm light irradiation, respectively. ROS detection was more evident for cells treated with AO at low concentrations. In accordance with the results obtained in MET1 cells, MB and AO were also cytotoxic and phototoxic towards the melanoma cell line; in particular, AO demonstrated a significant difference between the IC_50_ values of cells with and without irradiation. This investigation also revealed that although the molecules can act as photosensitizers, the effects are dependent on the cell type and the concentration of the photosensitizer. Finally, this work allowed demonstrates the significant photosensitizing activity of AO at low concentrations and under moderate light doses. Therefore, our aim in this work was achieved, i.e., to perform a broad study of molecules that have not been extensively explored for PDT, employing wide concentration ranges that can be helpful for futures studies, in particular, identifying the most promising dyes as methylene blue and acridine orange, in order to determine concentration ranges that are efficient for photodynamic therapy of cancer cells.

## Figures and Tables

**Figure 1 ijms-24-05602-f001:**
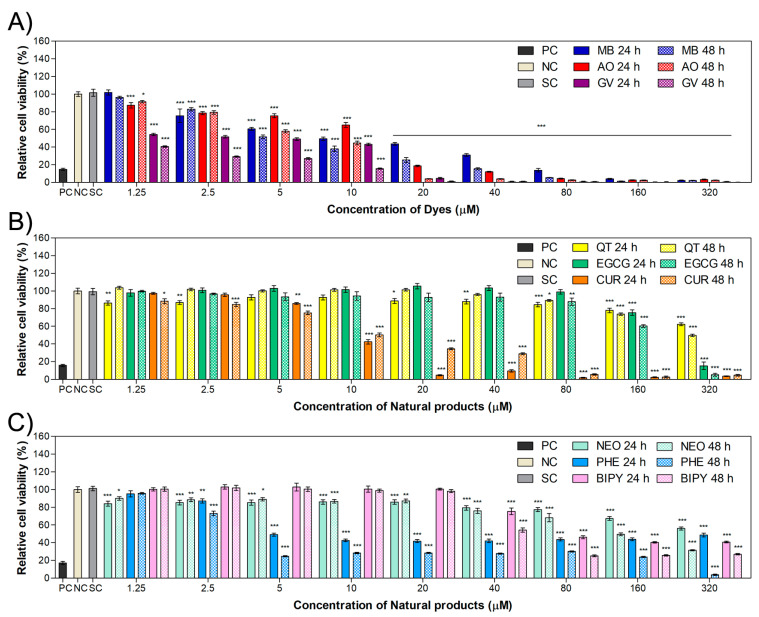
Cell viability of HaCaT cells treated for 24 or 48 h with (**A**) dyes (MB, AO and GV), (**B**) natural compounds (QT, CUR and EGCG) or (**C**) chelating molecules (NEO, PHE and BIPY) with a range of concentration from 1.25 µM to 320 µM. Values are presented as the mean ± combined standard uncertainty (*n* = 4), and statistical analysis comparing the results to the cell viability of the negative control was performed by a two-way ANOVA with Bonferroni post test, where * *p* < 0.05, ** *p* < 0.01 and *** *p* < 0.001. MB: methylene blue, AO: acridine orange, GV: gentian violet, QT: quercetin, EGCG: epigallocatechin gallate, CUR: curcumin, NEO: neocuproine, PHE: 1,10-phenanthroline, BIPY: 2,2′-bipyridyl, PC: positive control, NC: negative control, SC: solvent control.

**Figure 2 ijms-24-05602-f002:**
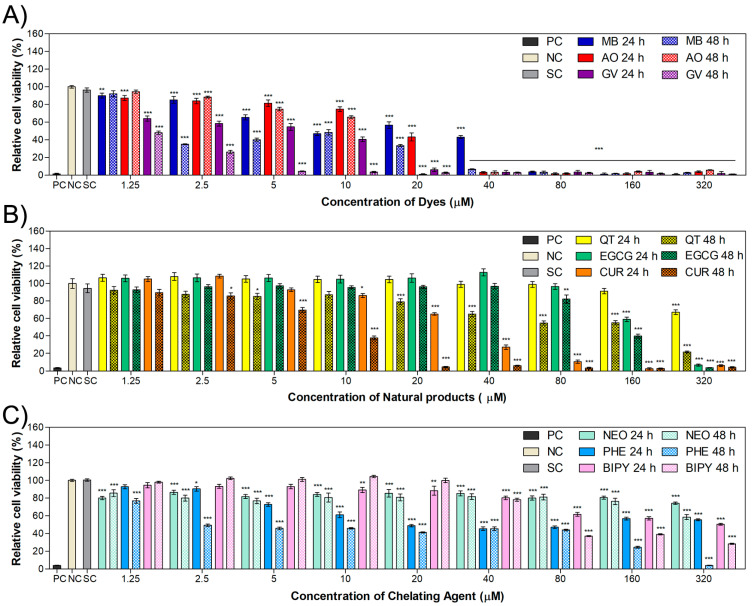
Relative cell viability of the MET1 cell line treated for 24 or 48 h with (**A**) dyes (MB, AO and GV), (**B**) natural compounds (QT, CUR and EGCG) or (**C**) chelating molecules (NEO, PHE and BIPY) with a range of concentration from 1.25 µM to 320 µM. Values are presented the mean ± combined standard uncertainty (*n* = 4), and statistical significances comparing the results to the cell viability of the negative control were determined by a two-way ANOVA with Bonferroni post test, where * *p* < 0.05, ** *p* < 0.01 and *** *p* < 0.001. MB: methylene blue, AO: acridine orange, GV: gentian violet, QT: quercetin, EGCG: epigallocatechin gallate, CUR: curcumin, NEO: neocuproine, PHE: 1,10-phenanthroline, BIPY: 2,2′-bipyridyl, PC: positive control, NC: negative control, SC: solvent control.

**Figure 3 ijms-24-05602-f003:**
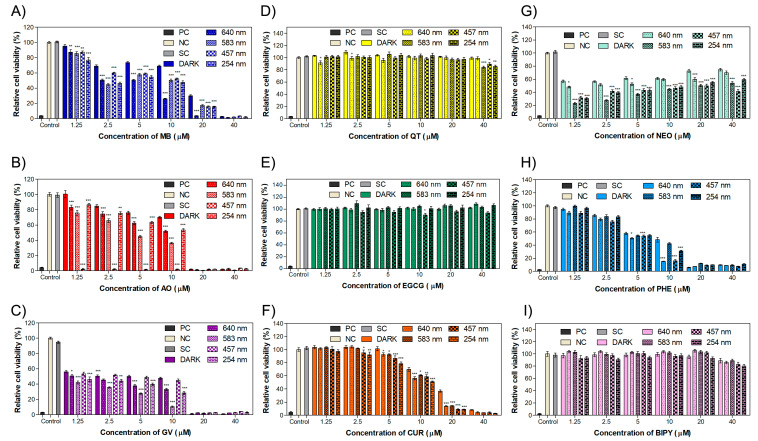
Relative cell viability of the MET1 cell line treated for 24 h with dyes (**A**) MB, (**B**) AO, (**C**) GV; natural compounds (**D**) QT, (**E**) EGCG, (**F**) CUR; or chelating agents (**G**) NEO, (**H**) PHE, (**I**) BIPY and submitted to irradiation. Samples kept in the dark were used as controls and compared to those submitted to irradiation at different wavelengths. Values are presented mean ± combined standard uncertainty (*n* = 6), and statistical analysis comparing the results to the dark control was performed with a two-way ANOVA with Bonferroni post test, where * *p* < 0.05, ** *p* < 0.01 and *** *p* < 0.001. MB: methylene blue, AO: acridine orange, GV: gentian violet, QT: quercetin, EGCG: epigallocatechin gallate, CUR: curcumin, NEO: neocuproine, PHE: 1,10-phenanthroline, BIPY: 2,2′-bipyridyl, PC: positive control, NC: negative control, SC: solvent control.

**Figure 4 ijms-24-05602-f004:**
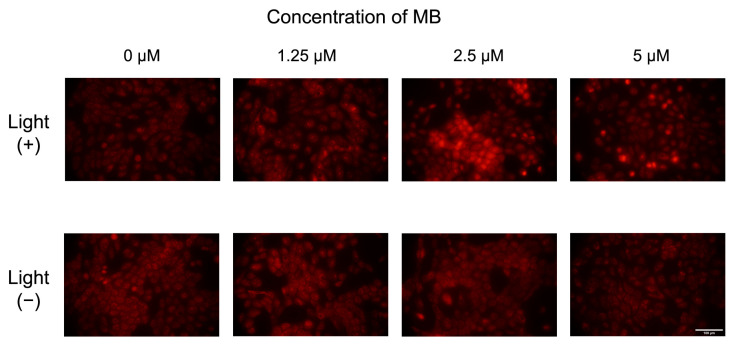
Fluorescence images of MET1 cells in the presence of different concentrations of methylene blue (MB) with the presence or absence of light (640 nm). The light dose applied to the irradiated cells was 2.5 J/cm². Scale bar = 100 µm.

**Figure 5 ijms-24-05602-f005:**
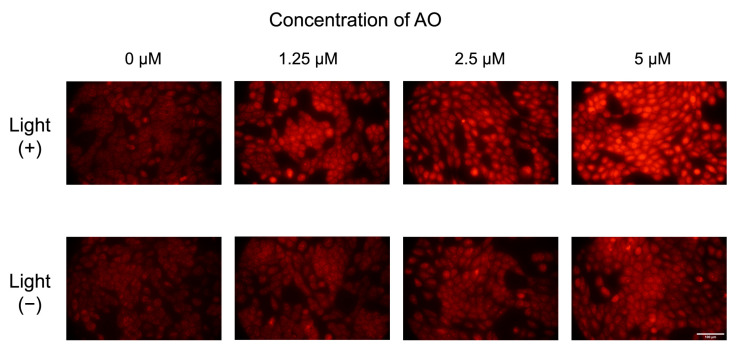
Fluorescence images of MET1 cells in the presence of different concentrations of acridine orange (AO) with the presence or absence of light (457 nm). The light dose applied to the irradiated cells was 2.5 J/cm². Scale bar = 100 µm.

**Figure 6 ijms-24-05602-f006:**
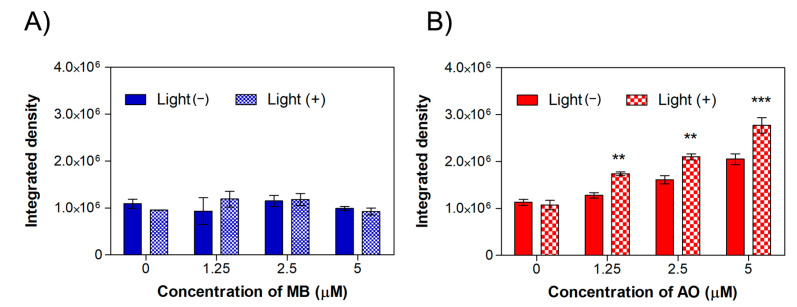
Quantification of fluorescence intensity of MET1 cells treated with (**A**) methylene blue and (**B**) acridine orange either submitted or not to irradiation at 640 nm and 457 nm, respectively. Results are displayed as mean ± standard deviation of the mean (*n* = 4), and statistical analysis was performed using a two-way ANOVA with Bonferroni post test, where ** *p* < 0.01 and *** *p* < 0.001, with comparison of the fluorescence intensity of irradiated cells with the cells treated with the same concentration and non-irradiated cells.

**Figure 7 ijms-24-05602-f007:**
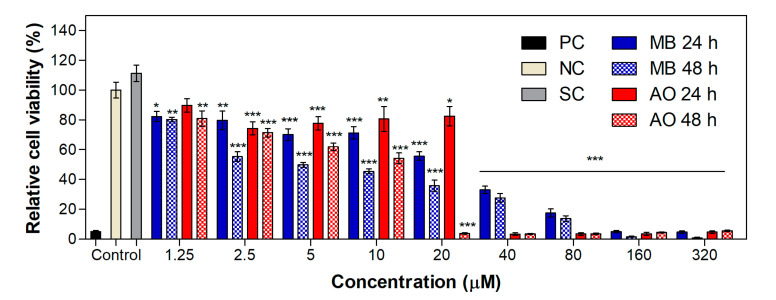
Relative cell viability of the WM983b cell line treated for 24 or 48 h with the dyes MB and AO within a range of concentration from 1.25 µM to 320 µM. Values are presented as the mean ± combined standard uncertainty (*n* = 4), and statistical analysis comparing the results to the negative control was performed with a two-way ANOVA with Bonferroni post-test, where * *p* < 0.05, ** *p* < 0.01 and *** *p* < 0.001. MB: methylene blue, AO: acridine orange, PC: positive control, NC: negative control, SC: solvent control.

**Figure 8 ijms-24-05602-f008:**
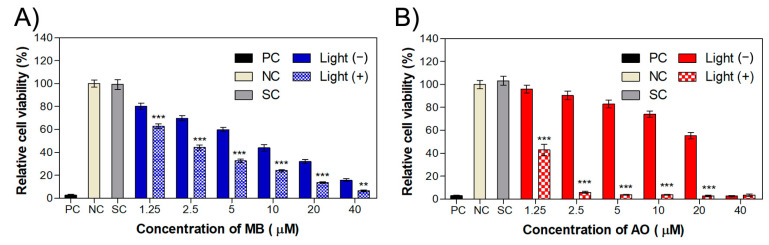
Relative cell population of the WM983b cell line treated for 24 h with dye molecules (**A**) MB and (**B**) AO and submitted to irradiation. Samples not treated with irradiated light (−) were used as controls for comparison with those submitted to irradiation at different light wavelengths (640 nm for MB and 457 nm for AO). The light dose applied to the irradiated cells was 2.5 J/cm². Values are presented as means ± combined standard uncertainty (*n* = 6), and statistical analysis was performed with a two-way ANOVA with Bonferroni post test, where ** *p* < 0.01 and *** *p* < 0.001, in comparison with the dark control. MB: methylene blue, AO: acridine orange, PC: positive control, NC: negative control, SC: solvent control.

**Table 1 ijms-24-05602-t001:** Calculated IC_50_ values in 24 and 48 h cytotoxicity tests using MET1 and HaCaT cells treated with the dyes (methylene blue, acridine orange and gentian violet), natural compounds (quercetin, curcumin and epigallocatechin gallate) and chelating molecules (neocuproine, phenanthroline and 2,2′-bipyridyl) using concentrations from 1.25 µM to 320 µM.

	IC_50_ (µM)			
	MET1 Cells		HaCaT Cells	
Molecule	24 h	48 h	24 h	48 h
Methylene blue	14.2	4.2	11.6	7.1
Acridine orange	15.1	10.3	10.5	6.5
Gentian violet	3.9	1.2	2.8	0.9
Quercetin	>320	110	>320	315
Curcumin	26.0	7.2	9.0	11.7
Epigallocatechin gallate	174	136	211	174
Neocuproine	>320	>320	>320	159
1,10-phenanthroline	124	7.7	35.2	5.9
2,2′-bipyridyl	275	97.2	125	57.8

**Table 2 ijms-24-05602-t002:** Comparison of IC_50_ values from a phototoxicity assay in MET1 cells treated with dyes (methylene blue, acridine orange and gentian violet), natural compounds (quercetin, curcumin and epigallocatechin gallate) and chelating molecules (neocuproine, phenanthroline and 2,2′-bipyridyl) exposed (or not, i.e., dark control) to a 2.5 J/cm² light dose at different wavelengths.

	IC_50_ (µM)				
Molecule	Dark	640 nm	583 nm	457 nm	254 nm
Methylene blue	10.9	3.8	5.0	6.2	4.3
Acridine orange	11.5	6.6	4.1	0.4	7.1
Gentian violet	2.9	1.8	1.0	2.7	1.5
Quercetin	230	340	269	288	205
Curcumin	15.5	11.0	11.5	10.7	9.3
Epigallocatechin gallate	191	224	211	222	193
Neocuproine	-	-	-	-	-
1,10-phenanthroline	7.4	4.8	6.8	4.9	6.0
2,2′-bipyridyl	228	202	190	171	225

## Data Availability

The data presented in this study are available upon request from the corresponding author. The data are not publicly available, as they are presented in the paper.

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
