# Peer review of "Phototoxic Potential of Different DNA Intercalators for Skin Cancer Therapy: In Vitro Screening"

_ijms, 2023, doi:10.3390/ijms24065602_

Round 1

Reviewer 1 Report

All suggestions are in below:

General comment

The article presents interesting findings on the phototoxicity of HaCaT and MET1 cells in relation to PDT reactions with PSs as well as melanoma cell lines. However, in order to ensure safe PDT treatment for patients, it is important for the article to provide a comprehensive discussion on the permeation of PSs into different types of cells. Additionally, the article could benefit for clinicians from the following revisions:

Specific Comment

Introduction:

Line 79-81

The range of concentrations of substances (e.g. curcumin and methylene blue) that have been shown to be toxic to certain types of cells should be mentioned.

Line 133-134

A more detailed explanation of the lower toxicity of natural products should be provided or postulated.

Line 167-179:

Please include the time point used in this part of the experiment.

Figure 3:

Please indicate the time point (24 or 48 hrs) that was used in these experiments in the figure legend.

Line 214-215:

It is unclear which concentrations of EGCG exert prooxidant effects and which concentrations of EGCG result in antioxidant effects. Please provide this crucial information.

Table 2:

The name of the table should include more information about the power density and energy density of the light.

Line 270-274:

The main products of PDT are hydroxyl radical and singlet oxygen, but the authors emphasize superoxide. It is recommended that the article discuss this point more or explain why the main probes for hydroxyl and singlet oxygen (e.g. APF  or (2-[6-(4amino phynoxyl-3H-xanthen-3-on-9-yl] benzoic acid) and SOGS (singlet oxygen sensor green) were not chosen.

Figure 4 and 5:

The legends of these figures should be adjusted to include the wavelength and energy density used.

Figure 8:

Please indicate the time point (24 or 48 hrs) that this part of the study was performed, as well as the power density and energy density used.

Materials and Methods:

The authors should specify the units of concentration (e.g. w/w, w/v, v/v, or v/w) when mentioning percentages (e.g. Line 396, 397, 415).

The article should mention all types of solvents used, as some PSs are not water-soluble.

Line 411-413

Pre-irradiation (drug-light interval time) should be included in Section 3.4.

Line 418

The article should mention not only the energy density but also the power density in Section 3.4. Also the dimension of the light bulb used.

Line 422

The word "section 2.3" should be changed to "Section 3.3" in line 422.

Section 3.5

The article should explain how the authors chose the areas to be measured under the fluorescent microscope without bias, including how many areas, under what magnification.

Author Response

Reviewer1:

General comment

The article presents interesting findings on the phototoxicity of HaCaT and MET1 cells in relation to PDT reactions with PSs as well as melanoma cell lines. However, in order to ensure safe PDT treatment for patients, it is important for the article to provide a comprehensive discussion on the permeation of PSs into different types of cells. Additionally, the article could benefit for clinicians from the following revisions:

Comment from the authors: we thank the reviewer 1 for their careful analysis of our manuscript. Answers to specific comments are detailed below. We hope manuscript improvements adequately reflect reviewer's suggestions.

Specific Comment

Introduction:

Comment 1: Line 79-81

The range of concentrations of substances (e.g. curcumin and methylene blue) that have been shown to be toxic to certain types of cells should be mentioned.

Answer from the authors: We understand that as non-conventional molecules these should be better described regarding the concentrations used and treatments performed on the literature cited where a PDT effect have been shown. Therefore, we added these details.

Comment 2: Line 133-134

A more detailed explanation of the lower toxicity of natural products should be provided or postulated.

Answer from the authors: An explanation related to the cytotoxicity of natural compounds to cancer cells was added to the manuscript.

Comment 3: Line 167-179:

Please include the time point used in this part of the experiment.

Answer from the authors: This information in provided in the materials and methods section but we understand it is important to write again the experimental time point therefore we added this information.

Comment 4: Figure 3:

Please indicate the time point (24 or 48 hrs) that was used in these experiments in the figure legend.

Answer from the authors: We added this information in the figure caption.

Comment 5: Line 214-215:

It is unclear which concentrations of EGCG exert prooxidant effects and which concentrations of EGCG result in antioxidant effects. Please provide this crucial information.

Answer from the authors: A more detailed explanation related to the dual role of EGCG regarding oxidant properties was added in the manuscript.

Comment 6: Table 2:

The name of the table should include more information about the power density and energy density of the light.

Answer from the authors: We added the information on the light dose because it is indeed relevant when we comment on cell irradiation.

Comment 7: Line 270-274:

The main products of PDT are hydroxyl radical and singlet oxygen, but the authors emphasize superoxide. It is recommended that the article discuss this point more or explain why the main probes for hydroxyl and singlet oxygen (e.g. APF  or (2-[6-(4′amino phynoxyl-3H-xanthen-3-on-9-yl] benzoic acid) and SOGS (singlet oxygen sensor green) were not chosen.

Answer from the authors: Due to the treatment with the dyes we had to be careful with the choice of the probe. We indeed had others at our disposal such as DCFDA for hydroxyl and peroxyl, however our dye acridine orange led to fluorescence interference and therefore we have chosen a dye with different excitation/emission wavelengths that would not be affect by our treatments as was the case of DHE. We made it also clear in the manuscript.

Comment 8: Figure 4 and 5:

The legends of these figures should be adjusted to include the wavelength and energy density used.

Answer from the authors: We appreciate the suggestion, and we added these information’s in the captions of figures 4 and 5.

Comment 9: Figure 8:

Please indicate the time point (24 or 48 hrs) that this part of the study was performed, as well as the power density and energy density used.

 Answer from the authors: We added the suggested information in the captions of the figures 4 and 5.

Comment 10: Materials and Methods:

The authors should specify the units of concentration (e.g. w/w, w/v, v/v, or v/w) when mentioning percentages (e.g. Line 396, 397, 415).

The article should mention all types of solvents used, as some PSs are not water-soluble.

Answer from the authors: We added the information of the unit of concentration. The solvent used was only DMSO as mentioned also in the materials and methods as well as the maximum concentration of DMSO applied in the treatments and their respective control employed in the experiments.

Comment 11: Line 411-413

Pre-irradiation (drug-light interval time) should be included in Section 3.4.

Answer from the authors: This information is described in the materials and methods, but we added also when introducing the description of the treatments to be clear that it was a 24 hour treatment pre-irradiation.

Comment 12: Line 418

The article should mention not only the energy density but also the power density in Section 3.4. Also the dimension of the light bulb used.

Answer from the authors: Thank you for the suggestion. This information was included in the materials and methods section.

Comment 13: Line 422

The word "section 2.3" should be changed to "Section 3.3" in line 422.

Answer from the authors: We corrected that reference to the section.

Comment 14: Section 3.5

The article should explain how the authors chose the areas to be measured under the fluorescent microscope without bias, including how many areas, under what magnification.

Answer from the authors: Thank you very much for this suggestion. We have added the information related to the acquisition of the fluorescence images in the materials and methods section 3.5.

Reviewer 2 Report

The manuscript is correctly written and describes a job well carried out, with results that support the conclusions.

However, the results do not represent a new contribution to pre-existing knowledge on the subject. The compounds reported have already been described as photosensitizers, as has also been described the variable effect of PDT depending on the cell line used.

To improve the manuscript, it could be proposed to broaden the spectrum of cell lines used, or at least expand the ¨Conclusions¨ section, in order to highlight what the novel contribution of the work would be for the authors.

Author Response

 Reviewer2:

The manuscript is correctly written and describes a job well carried out, with results that support the conclusions.

However, the results do not represent a new contribution to pre-existing knowledge on the subject. The compounds reported have already been described as photosensitizers, as has also been described the variable effect of PDT depending on the cell line used.

To improve the manuscript, it could be proposed to broaden the spectrum of cell lines used, or at least expand the ¨Conclusions¨ section, in order to highlight what the novel contribution of the work would be for the authors.

Answer from the authors: We thank you for the comments and the suggestions. Indeed, there are some studies with some of the molecules tested in our experiments however not much in the literature is explored as conventional photosensitizers porphyrins, phthalocyanines and other classical PS molecules. Our aim in this work was to study some molecules not much explored such as methylene blue and curcumin together with some other such as acridine orange or natural compound as quercetin that have only a few studies related to photodynamic therapy, sometimes not even directly as PS but as a synergistic drug. Therefore, it is a wide range experiment employing broad concentration ranges that can be helpful for futures studies with the molecules mainly those most promising as methylene blue and acridine orange to determine concentrations range that are efficiently resulting in the photodynamic therapy of cancer cells. This information was included in the conclusions.  

Reviewer 3 Report

The results in the manuscript are interesting. In Introduction: why havew you choosen these cell types? Please add some sentences aout the skin and melanoma diseases.

In the Results and Discussion section: Figurres are too small and should be enlarged. Have you performed ROS on HaCAT cells? There are missing references about that all the tested compounds are DNA-intercallators.

Can you relate the photophysical properties of the compounds (triplet state etc.) with the results of the cell experiments? How can you explain the obtained results knowing their photophysical characteristics?

In section Material and Methods: How cells were irradiated - from bottom or above? The device is not mentioned - what laser and brand? All the used devices should be described precisely. Have you covered the plates by alluminium folium for the PDT experiments? Please, explain deeply the results comparing different effects in %. Results and discussion should be better written.

Author Response

Reviewer3:

Comment 1: The results in the manuscript are interesting. In Introduction: why havew you choosen these cell types? Please add some sentences aout the skin and melanoma diseases.

Answer from the authors: We added some information regarding PDT for skin cancers.

Comment 2: In the Results and Discussion section: Figurres are too small and should be enlarged. Have you performed ROS on HaCAT cells? There are missing references about that all the tested compounds are DNA-intercallators.

Answer from the authors: We made alterations and achieved the maximum dimensions of figures for the IJMS manuscript template. Regarding to the HaCaT question we wanted to see the possible cytotoxicity of the compounds on non-tumor cell line however we have not performed ROS on HaCaT cells. Considering that photodynamic therapy is a local therapy it should not influence healthy and normal cells. And we have added the references regarding the intercalation properties of these molecules.

Comment 3: Can you relate the photophysical properties of the compounds (triplet state etc.) with the results of the cell experiments? How can you explain the obtained results knowing their photophysical characteristics?

Answer from the authors: It is a very interesting point and in fact mostly for the dyes that showed the most promising results of phototoxicity at low concentrations, the photophysical properties have for sure influence on the results. For example, the results for acridine orange agree with the respective spectrum, such as the band around 266 nm and the band in the visible region at 490 nm wavelength (maximum absorbance) due to AO molecule (monomer) and a shoulder at 470 nm (possibly due to AO dimers). In the case of AO spectra there are two main peaks one around 488 nm and another at 266 nm, both corresponding to π‒π* transitions. Irradiation targeting these wavelengths showed to have a significant reduction on the IC50 compared to the dark control.

Comment 4: In section Material and Methods: How cells were irradiated - from bottom or above? The device is not mentioned - what laser and brand? All the used devices should be described precisely. Have you covered the plates by alluminium folium for the PDT experiments? Please, explain deeply the results comparing different effects in %. Results and discussion should be better written.

Answer from the authors: We added the information that cells were irradiated from above and we have mentioned the light sources and the laboratory-made irradiation systems. Regarding the aluminum foil, we used an incubator dedicated to this experiment that ensured the experiment was for sure not affected by light coming from other sources such as ambient light. We also made some changes to make the most important results obtained clear.

Round 2

Reviewer 2 Report

-